# The Comparison of Nutritional Value of Human Milk with Other Mammals’ Milk

**DOI:** 10.3390/nu12051404

**Published:** 2020-05-14

**Authors:** Renata Pietrzak-Fiećko, Anna M. Kamelska-Sadowska

**Affiliations:** 1Department of Commodities and Food Analysis, Faculty of Food Sciences, University of Warmia and Mazury in Olsztyn, 1 Cieszyński Square, 10-726 Olsztyn, Poland; renap@uwm.edu.pl; 2Department of Rehabilitation and Orthopedics, Collegium Medicum, University of Warmia and Mazury in Olsztyn, 2 Oczapowskiego Street, 10-719 Olsztyn, Poland; 3Clinic of Rehabilitation, Provincial Specialist Children’s Hospital in Olsztyn, 18A Żołnierska Street, 10-561 Olsztyn, Poland

**Keywords:** cholesterol, fat, fatty acids, human, mare, cow, sheep, goat, milk, nutrients

## Abstract

(1) Background: The variation in the concentration of different components found in milk depends on mammalian species, genetic, physiological, nutritional factors, and environmental conditions. Here, we analyse, for the first time, the content of different components (cholesterol concentration and fatty acids composition as well as the overall fat and mineral content determined using the same analytical methods) in milk of different mammal species. (2) Methods: The samples (*n* = 52) of human, cow, sheep, goat and mare milk were analyzed in triplicate for: cholesterol concentration, fatty acids profile and fat and mineral content (calcium, magnesium, sodium, potassium, iron, zinc). (3) Results: The highest fat content was reported in sheep milk (7.10 ± 3.21 g/dL). The highest cholesterol concentration was observed in bovine (20.58 ± 4.21 mg/dL) and sheep milk (17.07 ± 1.18 mg/dL). The saturated fatty acids were the lowest in human milk (46.60 ± 7.88% of total fatty acids). Goat milk had the highest zinc (0.69 ± 0.17 mg/dL), magnesium (17.30 ± 2.70 mg/dL) and potassium (183.60 ± 17.20 mg/dL) content. Sheep milk had the highest sodium (52.10 ± 3.20 mg/dL) and calcium (181.70 ± 17.20 mg/dL) concentration values. (4) Conclusions: The differences in nutritional value of milk could be perceived as a milk profile marker, helping to choose the best food for human nutrition.

## 1. Introduction

Milk is composed of the following main components: water, fat, lactose, whey proteins, and minerals (ash) in amounts depending on the species of animals [1]. Milk composition is highly variable and differs even in the following days of lactation. Moreover, genetic, physiological, and nutritional factors, as well as environmental conditions, play a great role in making these differences [2,3].

Recently, the use of colostrum and mature milk derived from various mammals in human feeding has been gaining more and more popularity [4]. The reason for this is mainly its hypocholesterolemic action, better bioavailability, therapeutic properties (used in gastro-intestinal disorders) and no allergies after consumption. Milk fat is also one of the few food sources of butyric acid, a potent inhibitor of cancer cell proliferation as well as an inducer of differentiation and apoptosis in a number of cancer cell lines [5].

### 1.1. The Level of Fat in Milk and its Significance

Fat is the most variable component in milk. It varies during lactation, during the day and depends on the individual characteristics of breastfeeding women or feeding animals [6,7,8].

The mean fat content reported in our previous study in mature human milk was 3.8% (ranging from 1.63% to 6.40%) [7]. This value was same as reported by Thakore et al. (2018) (3.8%) on the 44th day of lactation and similar to those reported by Butts et al. (2018) in a New Zealand European mother population (3.72%) [9,10]. It was shown previously that the level of fat assessed in pre-term mothers was 30% higher than in term mothers [11].

The fat content reported in human (3.64%) and cow’s milk (3.61%) was higher compared to mare’s milk (1.21%) [12]. In milk samples analysed by Malacarne et al. (2002), sheep milk presented the highest fat content, reaching 6.9 ± 1% [12]. The fat content reported in goat milk was 4.2 ± 0.9% [13]. The highest value of fat was reported in hooded seal (61%) and reindeer milk (22.46%) [1,14].

The fat content in buffalo milk on the 3rd day of lactation (6.5%) was higher than milk from other species, while fat content was lower in cow milk on the 150th day of lactation (2.5%) than in human, goat, and buffalo milk [9]. 

From ruminants, the greatest fat content was reported in milk from buffalos (maximal value 15.0%) and the lowest in milk from cows (minimal value 3.3%). The low-fat content was found in milk from donkeys and mares. The fat content of women’s milk (2.1–4.0%) was greater than that of non-ruminants, but less than that of ruminants [8]. The overall fat content in different species according to selected authors is shown in Table 1.

An infant absorbs 92% of the breast milk lipids. Fat from breast milk is perceived as the main source of energy which covers the energy needs by 50% [19]. Moreover, fats are the tissue building materials. Fatty acids [specifically linoleic acid (LA)] are the precursors of biologically active substances (e.g., eicozanoids), in particular prostaglandins classified as tissue hormones such as prostacyclins (PGI_2_) [20,21,22,23].

In milk, 98–99% of fats are triacylglycerols. The remaining 1–2% are phospholipids, cholesterol, carotenoids, fat-soluble vitamins, some fatty acids and cholesterol esters [24,25].

### 1.2. The Cholesterol Concentration in Milk and Its Significance

The cholesterol concentration (ChC) assessed in mammals’ milk using different analytical methods is shown for comparison in Table 2. It was shown previously that the analytical method used for the determination of cholesterol in milk has an influence on the assessed ChC value [26].

The mean cholesterol concentrations (ChC) in our previous study in human milk collected at different stages of lactation (for women in 1–3, 5–6 and >6 month of lactation) were: 11.06 ± 3.51 mg/dL (4.94–14.63 mg/dL), 9.37 ± 3.46 mg/dL (4.30–14.92 mg/dL), 10.03 ± 3.88 mg/dL (4.68–21.77 mg/dL), respectively. These values were obtained using attenuated total reflectance Fourier transformed spectroscopy (FTIR-ATR). The ChC obtained from gas chromatography method ranged from 6.43 mg/dL to 28.19 mg/dL [7]. 

It was shown that the cholesterol concentration was similar in women and ruminants’ milk, but lower in that of non-ruminants [8].

It was presented previously that milk and dairy products showing high levels of fat, saturated fatty acids as well as cholesterol were associated with increased risk of cardiovascular disease (CVD). It was also assumed that dietary saturated fatty acid (SFA) may lead to increased low-density lipoprotein cholesterol (LDL) levels which, in turn, could lead to an increased risk of cardiovascular disease [43].

However, some studies showed that the subclinical inflammation was recognized as a key feature of the atherosclerotic process [44], and several markers of subclinical inflammation, such as C-reactive protein (CRP) and interleukin-6 (IL-6), were associated with an increased risk of coronary artery disease [45,46]. Vascular dysfunction [47] and impaired coagulation processes [48,49], as well as insulin resistance [50], were identified as key cardiometabolic disease risk factors.

Previous studies also suggested that high levels of cholesterol that appear in milk have a protective role for infants and program the metabolism of cholesterol in later life [51]. Therefore, mothers are advised to avoid the use of infant formulas which are said to have low cholesterol levels [52].

### 1.3. Fatty Acids Composition in Mammals’ Milk

Fatty acids (FAs) are essential for the proper development of a newborn. The fatty acids in milk come from the mother’s diet, resources of fats or they are synthesized in the liver and mammary gland [53], therefore their value could depend on the socioeconomic status of women [54].

Breast milk contains both long-chain polyunsaturated fatty acids (LCP) and essential fatty acids (EFAs). Colostrum contains high levels of arachidonic acid [ARA; 20:4 (*n*-6)] (more than 1% of total fatty acids) and docosahexaenoic acid [DHA; 22:6 (*n*-3)] (above 0.50%) [55,56].

In a baby’s diet, both linoleic acid [LA; C18:2 (*n*-6)] and linolenic acid [ALA; C18:3 (*n*-3)] are necessary. During the first year, the level of linoleic acid [(18:2 (*n*-6)] and the value of the alpha-linolenic acid [ALA; 18:3 (*n*-3)] do not show major changes.

Arachidonic acid (ARA) is a precursor of eicosanoids including leukotrienes (mediators of the immune response), prostaglandins (intracellular effectors), thromboxanes (which regulate vascular tone) and endocannabinoids [57,58]. ARA is the most predominant long-chain polyunsaturated fatty acid in human milk, albeit at low concentrations compared to other FAs. It was shown that ARA plays an important role in physiological development and its related functions during early life nutrition. Therefore, ARA is an important nutrient during infancy and childhood and, as such, appropriate attention is required regarding its nutritional status and presence in the infant diet [58].

Docosahexaenoic acid (DHA) is essential for the proper functioning of the retina in both preterm [59] and term [60] infants. The increase in DHA accretion in the third trimester coincides with brain and retina maturation [61].

Saturated (SFAs) and monounsaturated fatty acids (MUFAs) provide a large dose of energy to the young organism. This is due to the high content of palmitic acid in milk, constituting 50.8% of all SFA which corresponds to its large potential energy [62].

Saturated fats have been previously perceived as the maker for the development of CVD. However, recent findings have indicated that the link between SFA and CVD may be less clear than previously assumed. Foods are composed of an array of saturated and unsaturated fatty acids, each of which may differentially affect lipoprotein metabolism, as well as contribute significant quantities of other nutrients that may alter CVD risk [63,64].

The sum of saturated fatty acids in breast milk of European women was 44.9% of all fatty acids [65] compared to those living in areas of Kuwait, where the levels were 43% [53]. Pytasz (1999) studies showed that the level of saturated fatty acids was between 45–50% of the FA, including fatty acids from C6:0–C12:0 which were 6–8%, and the sum of myristic acid C14:0 and C16 palmitic acid represented 40% of all fatty acids in human milk [66]. 

The percentage of saturated fatty acids reported in cow milk were higher (65.6%) than in human milk (48.2%). On the other hand, the mono- and polyunsaturated FA were higher in human milk than in cow milk (39.8% and 10.8% vs. 30.3% and 4.5%) [67].

### 1.4. Minerals in Human Milk

The main minerals found in human milk are calcium (Ca), phosphorus (P), magnesium (Mg), potassium (K), sodium (Na) and chloride (Cl). The most common macrominerals assessed in milk of mammals are Ca, Mg, P, K, and Na; microminerals were e.g., copper (Cu), iron (Fe), manganese (Mn), molybdenum (Mo) and zinc (Zn), and heavy metals were aluminium (Al), arsenic (As), cadium (Cd), and lead (Pb) [68].

It was shown that human milk had lower values of Na, K, Ca, Mg, P, Cl, Zn in comparison with cow milk (15, 58, 34, 3, 15, 42, 1–3 mg/100 g vs. 43, 156, 120, 11, 94, 87, 30, 4 mg/100 g, respectively) and higher values of Cu (0.2–0.4 µg/mL vs. 0.05–0.2 µg/mL, respectively) [67].

The level of sodium was different during lactation and decreased in the days following human lactation. Major changes were observed in the case of transitional and mature milk. Lowering the concentration of these components was preceded by the decline in the infant needs [69]. The level of electrolytes in breast milk showed a constant value during the first four months of lactation for sodium: 141 ± 17 mg/L, potassium 480 ± 11 mg/L and chloride 452 ± 32 mg/L [70].

The level of zinc reported in human milk also decreased following days of lactation (from 8.18 mg/L at two days after delivery to 0.93 mg/L at 180 days after delivery), regardless of environmental factors or diet and had a value below the recommended standard. In the first days after delivery the level of zinc in milk decreases drastically and continues to decrease until about six weeks, setting a constant level around the third month [71]. It was shown that the level of zinc and nitrogen is the first limiting factor influencing the infant’s weight gain [72]. However, no correlation was found between the unique distribution of the concentration of zinc in relation to the needs of infants. It was only proven that a zinc deficiency in breastfed babies, especially those born preterm, is associated with the occurrence of dermatitis at the age of three months [71].

It was previously shown that buffalo milk had the highest Ca (204.23 ± 7.98 mg/100 g), P (117.45 ± 5.26 mg/100 g) and Mg (23.53 ± 1.33 mg/100 g) and the lowest Na (42.39 ± 0.82 mg/100 g) contents compared to goat and cow milk. Goat milk had the highest level of K (174.85 ± 4.85 mg/100 g). K was also the major mineral in cow and goat milk, while Ca was the major mineral in buffalo milk [73].

Not many of the previously mentioned scientists assessed the nutritional value of mammals’ milk by taking different groups of chemical compounds found in milk as well as the lipid quality indices in one article. This study compared different milk components with respect to selected animal species. Thus, the aim of the study was to compare the nutritional value of selected components (fat content, cholesterol concentration, fatty acids profile and mineral content as well as lipid quality indices) of different mammals’ milk determined by using the same analytical methods.

## 2. Materials and Methods 

### 2.1. Milk Samples

A total of 52 milk samples of human (*Homo sapiens*) (*n* = 18), cow (*Bos taurus*) (Holstein-Frisian breed; *n* = 10), sheep (*Ovis aries*) (Kamieniecka breed; *n* = 6), goat (*Capra hircus*) (White goat breed; *n* = 6) and mare (*Equus cabalus*) (Polish cold-blooded horse; *n* = 12) were included in this study. All animals were 5 years old. Human milk was collected from women aged 21–34 years old living in Olsztyn, in the third–fourth month of lactation. Other mammals’ milk samples were collected from small individual farms in the Warmia and Mazury region, also in the third–fourth month of lactation.

### 2.2. Experimental Design and Research Methods

#### 2.2.1. Ethical Considerations

The study protocol conforms to the ethical guidelines of the 1975 Declaration of Helsinki as reflected in a priori approval by the institution’s human research committee (the Ethics Committee of the Medical University of Bialystok No. R-I-003/22/2000). The experiment was conducted with the understanding of each participant. All lactating mothers gave written informed consent to participate in this study. All animals were used in compliance with the national laws and regulations of our research institutions.

#### 2.2.2. Experimental Design

Samples of human milk were collected at subjects’ houses. Milk was expressed immediately after feeding the baby (hind milk, mature milk) with the use of a breast pump. This specific way of taking samples arose from the principle that the fat content increases as the breast is drained of milk during a feed. Therefore, sampling pre-feed human milk will give a lower total fat content than mid- or post-feed samples [74]. The milk was drawn out from one breast at the same time of the day. In some cases, the total pool from one day was collected and the samples were stored in sealed, sterile containers at −20 ± 1 °C for no longer than 48 h.

All the animals of each species were kept in identical environmental conditions. All samples of mammals’ milk were obtained in the same region, which is known as the ecological area called “The lungs of Poland” and “Land of thousand lakes”. 

The cows were milked twice a day with the use of a pipeline milking machine. The samples of other mammals’ milk was obtained by hand milking by qualified staff. When milking, all hygiene procedures were followed and the samples were put into sterile containers.

The milk samples were sent (within no more than 48 h) on ice to the University of Warmia and Mazury in Olsztyn, Poland.

Before analysis, the milk samples were stored in a freezer at a temperature of −20 ± 1 °C. All samples were analysed in triplicate.

#### 2.2.3. Lipid Extraction and Milk Fat Content Determination

The tested milk was heated in a water bath at 40 ± 1 °C. Then it was stirred without causing excessive foaming or agitation and divided into equal parts. 

Milk fat content was determined by Röse-Gottlieb method (Association of Official Analytical Chemists, AOAC, 1990; PN-EN ISO 1211: 2011) [75]. The extraction procedure was performed in triplicate. The collected fractions were evaporated on a rotary vacuum evaporator.

#### 2.2.4. Sample Pre-Treatment for Cholesterol Concentration Quantification by IDF Standard Method

The concentration of cholesterol in raw mature milk was determined by the International Dairy Federation (IDF) standard method (1992) and gas chromatography (GC). The influence of direct saponification duration (30 min and one hour at the same temperature of 60 °C) on cholesterol concentration in milk was tested in a previous study [32]. Therefore, in this study the saponification process was carried out at 60 ± 1 °C for one hour. The extract used for chromatographic analysis was obtained as in Kamelska et al. (2011) [76]. The cholesterol content was analyzed using a PU-4600 chromatograph (Unicam, UK) equipped with a flame ionization detector (FID). The analysis was performed on a glass column (1 m × 4 mm) coated with a Chromosorb W (HP) 80/100 mesh. Argon was used as a carrier gas, flow rate was 50 cm^3^/min, and the detector and the injector temperatures were 300 °C and 290 °C, respectively.

#### 2.2.5. Fatty Acids Profile Determination

All types of fatty acids were converted into the corresponding fatty acid methyl esters (FAME) according to the IDF standard method (1999) using a methanolic solution of potassium hydroxide (KOH) (ISO 15884:2002) [77]. N-hexane and 2M KOH in methanol were added to the fat sample and the mixture was shaken. Then the sodium hydrogen sulphate (NaHSO_4_) was added and the mixture was centrifuged (3000 min^−1^). The methyl esters obtained in the process were then analysed by gas chromatography (GC). Chromatographic separation was performed using a Hewlett-Packard 6890 gas chromatograph with a flame-ionisation detector (FID) and a capillary column with a length of 100 m and internal diameter 0.25 mm. The liquid phase was Supelcowax 10 (Supelco, part of Sigma-Aldrich, St. Louis, MO, USA) and the film thickness was 0.25 µm. The conditions of separation were as follows for the carrier gas: helium, 1.5 mL/min flow rate, column temperature −60 °C, 5 °C/min increase to 180 °C, and the detector temperature −250 °C. Methyl esters of fatty acids were identified according to their retention times, which were compared with those of the mixture of methyl esters of fatty acids in the standard Supelco 37 Component FAME Mix (10 mg/mL in methylene chloride). For the calculation of the percentage share of fatty acids, the Chemostation computer programme (Agilent, Alpharetta, GA, USA) was used. Amounts of fatty acids were expressed as a weight percentage of total methyl esters of fatty acid.

#### 2.2.6. The Lipid Quality Indices

The Lipid Quality Indices were calculated according to the fatty acid composition using the following formulae (Ulbricht et al. 1991 [78]; Osmari et al. 2011 [79]):

Index of Atherogenicity (AI):AI = (C12:0 + (4 × C14:0) + C16:0)/(*n*-3PUFA + *n*-6PUFA + MUFA).(1)

Index of Thrombogenicity (TI)
TI = (C14:0 + C16:0 + C18:0)/((0.5 × C18:1) + (0.5 × sum of other MUFA) + 0.5 × *n*-6 PUFA) + (3 × *n*-3 PUFA) + *n*-3 PUFA/*n*-6 PUFA))(2)

Hypocholesterolaemic/hypercholesterolaemic ratio (HH) was calculated according to Santos-Silva et al. (2002) [80]
HH = [(sum of C18:1 *cis* 9, C18:2 *n*-6 and C18:3 *n*-3)/(sum of C14:0 and C16:0)].(3)

#### 2.2.7. Mineral Analysis

Six minerals including microelements such as iron (Fe) and zinc (Zn) as well as macroelements such as calcium (Ca), magnesium (Mg), sodium (Na) and potassium (K) were analysed in milk using flame atomic absorption spectrometry.

The samples of milk were dried in an oven at 105 °C to achieve a constant weight. Afterwards, the samples were carbonized and incinerated in 480 °C over several hours. Then the muffle-furnace ashing using a crucible in a muffle furnace (with the temperature raising gradually from 100 °C to 500 °C) was performed to obtain a white-grey ash. The ashes were heat-dissociated in 1M HNO_3_ (Merck, Darmstadt, Germany) and transferred to 25 mL volumetric flasks and filled up with de-ionized water (>18.2 MΩcm). Reagent samples were prepared in parallel to the test samples. The mineral content was determined in the obtained mineralisations (directly or after their dilution). To determine the Ca content, the addition of a 10% heptahydrated lanthanum chloride LaCl_3_*7H_2_O (Merck, Germany) in an amount providing a final concentration of La^+3^ of 1% was used.

The content of minerals (Ca, Mg, Na, K, Fe, Zn) was determined by flame atomic absorption spectrometry (flame:acetylene:air) using an iCE 3000 Series atomic absorption spectrometer (ThermoScientific, United Kingdom), equipped with a GLITE data station, background correction system (deuterium lamp) and appropriate cathode lamps [81]. Standard solutions of minerals were prepared immediately before use by dilution (with 0.1M HNO_3_, Merck, Germany) of standards at the concentration of 1 mg/cm (Baker, The Netherlands).

#### 2.2.8. Statistical Analyses

The statistical analyses were done using the Statistica 13.3 program (StatSoft, Inc., 2300 East 14th Street, Tulsa, OK 74104, USA). The normality of distribution was determined using the Shapiro-Wilk test. One-way analysis of variance (ANOVA) was used to assess the differences in the research component values between milk from different species. If the sample distribution and the normal distribution were non-compliant, testing was based on a non-parametric Wilcoxon test. The post-hoc Tukey test was used to find means that were significantly different from each other. Statistical significance was set at *p* < 0.05.

## 3. Results and Discussion

### 3.1. Fat Content and its Significance in Nutrition

The fat content, cholesterol concentration and fatty acid profile reported in selected mammals’ milk are shown in Table 3. 

The mean fat content reported in this study in mature human milk was 3.53% (Table 3). This value was similar to those reported by Malacarne et al. in 2002 [12].

The highest fat content was reported in sheep milk (7.10 ± 3.21 g/dL) and was statistically different from cow (2.90 ± 0.19 g/dL) and mare milk (1.21 ± 0.85 g/dL) (Table 3). The highest value of fat in sheep milk deemed it the best energy source. Previous studies confirmed our results—the fat content in mare’s milk was lower in comparison with human and cow milk [12].

It was shown previously that goat milk fat is also a good source of energy because it is easier to digest as the fat molecules in the goat milk are much smaller than the fat molecules in cow’s milk. Thus, it is perceived as a good fat source for human nutrition [82].

### 3.2. Nutritional Value and Health-Promoting Properties of Cholesterol Concentration in Milk

In this study, human milk presented low levels of cholesterol concentration (ChC) (9.90 ± 6.51 mg/dL). The ChC obtained from human milk in previous studies was in the broad range (4.30–28.19 mg/dL) [7]. Thus, it could be concluded that this value is highly variable and depends on many factors including the individual characteristics of women and the analytical method used for its determination [26]. Thus, in this study the same analytical method for the cholesterol determination was used.

The highest cholesterol concentration was observed in bovine milk (20.58 ± 4.21 mg/dL) and sheep milk (17.07 ± 1.18 mg/dL) and was much higher than that reported in human, goat and mare milk. It was previously shown that the high level of cholesterol found in milk is the main source for hormone production and has a positive influence on cholesterol metabolism in later life [51]. Thus, sheep milk as well as bovine milk could be a good replacement source of cholesterol in human nutrition when problems with natural breastfeeding occur.

Moreover, the newest data suggest that milk cholesterol concentration in mice is not affected by a high-cholesterol diet and conditions of maternal hypercholesterolaemia and is maintained at stable levels via ATP-binding cassette transporter G8 (ABCG8-) and LDL receptor (LDLR-) independent mechanisms [83]. The study in humans also showed that maternal diet does not change the ChC in breast milk [84].

Mare’s milk reported in this study had the lowest cholesterol concentration (6.30 ± 1.08 mg/dL). This value was statistically different from other mammals (*p* < 0.01) except for human milk. Cholesterol concentration in human milk was significantly lower than in cow milk and sheep milk. However, it should be again noted here that some individual characteristics of the women population could be associated with low determined levels of this component in that probe.

The ChC levels in this study were comparable with most of the other research. The cholesterol concentration reported in bovine milk in this study (20.58 ± 4.21 mg/dL) was similar to those assessed by Šterna and Jemeljanovs (2003) (16.25–18.63 mg/dL) [85]. 

The cholesterol values reported in sheep’s milk in this study were higher than in the study by Mayer and Fiechter (2012) [86]. 

The concentration of cholesterol in mature goat milk was lower (11.64 ± 1.09 mg/dL) than those reported by Strzałkowska et al. (2006) and Zaharia et al. (2011). It was also shown that ChC in colostrum (9.43 mg/dL) was lower than in mature milk (15.68–19.10 mg/dL) [87,88]. 

The results obtained in this study were also lower than those indicated by Gorban and Izzeldin (1999) [89]. The ChC determined by those authors were for human, goat, cow and sheep milk: 20.00 mg/dL, 13.00 mg/dL, 25.63 mg/dL, and 23.00 mg/dL, respectively.

### 3.3. The Comparison of Health-Promoting Properties of Fatty Acids in Selected Mammals’ Milk

ChC as well as fatty acid content was different in various mammalian species [90,91,92]. From the health point of view, the saturated fatty acids (SFAs) are perceived to increase the low-density lipoprotein cholesterol (LDL-C) and increase the risk of cardiovascular disease [93]. The sum of saturated fatty acids (SFA) was the lowest in human milk [46.60 ± 7.88% of total fatty acids (tFA)] and the highest in sheep milk (77.50 ± 0.92% of tFA). The lowest value of SFA in human milk (43.50%) was also reported by Strzałkowska et. al. (2012). The highest value was assessed in sheep milk (66.00%) [94]. 

The amount of free fatty acids and phospholipids reported by Gantner et al. (2015) were notably lower in milk from ruminants and women compared to milk from mares and donkeys [8]. The percentage of saturated and monounsaturated fatty acids was lower, while the unsaturated fatty acid content was higher in milk from non-ruminants, with a remarkably higher percentage of C-18:2 and C-18:3. This study showed that from the nutritional point of view, milk from non-ruminants could be more suitable for human nutrition than milk from ruminants [8]. 

The lowest level of short-chain fatty acids (with aliphatic tails of 6 to 12 carbons) was reported in human milk and the highest value was assessed in goat milk. Other authors confirmed the results of this study and the high levels of caproic acid (C6:0), caprylic acid (C8:0) as well as capric acid (decanoic acid; C10:0) found in goat milk [95].

The influence of fatty acid derived from diet on cholesterol concentration and cardiovascular disease is still confusing. It was previously shown that only 14% of fatty acids (FAs) derived from milk could increase the cholesterol concentration in blood serum. A total of 45% of FAs lower ChC and 41% is perceived to be neutral [96]. Moreover, feeding rats with lyophilized goat milk led to a decrease in ChC as well as triacylglycerol concentration in blood serum [97]. It was shown to be associated with the high level of medium-chain FAs. It should be noted that this leads also to a decrease in endogenic synthesis of cholesterol and its absorption in the small intestine. In this study, the goat milk presented a high level of medium-chain FAs (Table 3). It could be concluded that milk of this mammal could be used as a therapeutic trait in case of hypocholesterolaemic action.

When considering health-promoting properties of milk, the most important FAs are polyunsaturated FA (PUFA). The highest biological activity was shown for linoleic acid (CLA) [98]. From the nutritional point of view, higher percentage of unsaturated FA is significant. In this study, the lowest values of mono- and poly-unsaturated fatty acids were reported in sheep milk (19.01 ± 1.35% vs. 3.86 ± 0.49%) and goat milk (21.83 ± 0.52% vs. 2.97 ± 0.33%), respectively. The CLA level was higher in cow and sheep milk than in other examined mammals’ milk. 

It was shown previously that the saturated and unsaturated fatty acids content in milk was different depending on the breed and stage of lactation. In the milk of the Konik Polski breed, saturated fatty acids appeared predominate in the early and mid-lactation (51.95% vs. 52.95%), whereas unsaturated fatty acids (62.28%) predominated in the late lactation period. In the case of milk samples of the Polish cold-blooded mares, saturated fatty acids content in the early and late lactation was 55.77% vs. 61.31%, whereas unsaturated fatty acids in the mid-lactation were 52.20% [99].

It should be noted that long-chain derivatives e.g., arachidonic acid (*n*-6), eicosapentaenoic acid (20:5 *n-3*, EPA), and docosahexaenoic acid (C22:6 *n*-3, DHA) are produced from the sources of PUFA e.g., from linoleic acid (18:2 *n*-6, LA) and linolenic acid (C18:3 *n*-3, ALA). Mare milk presented the highest level of 18:2 fatty acids (14.94%). In comparison, in human milk the content of 18:2 FAs was 8.84% and in other mammal’s milk samples it was below 3.00%. The highest value of C18:3 *n*-3 was reported in mare milk (7.05%). In other mammals milk this value was below 1.00%. It was shown in other authors’ research that human milk had 15.24–17.73% of C18:2 FA and 0.6–1.36 % of C18:3 FA [4].

### 3.4. Lipid Quality Indices

The calculated Lipid Quality Indices are shown in Table 4. The highest index of atherogenicity (AI) was reported in sheep milk (4.21 ± 0.23) and goat milk (3.17 ± 0.19). The highest index of thrombogenicity (TI) was from milk derived from sheep (2.30 ± 0.13). The latter value was lower than that reported by Paszczyk et al. (2019) (3.03 ± 0.13) [17].

In Paszczyk et al.’s (2019) study, cow milk presented the highest index of atherogenicity (3.31 ± 0.21) and of thrombogenicity (3.59 ± 0.19). These values were higher than reported in this study. Goat and sheep milk from Paszczyk et al.’s studies had the highest hypocholesterolaemic/hypercholesterolaemic (HH) ratio (0.58 ± 0.01 and 0.56 ± 0.01, respectively) [17]. The hypocholesterolaemic/hypercholesterolaemic (H/H) ratio of raw milk from two sheep breed (Karagouniko and Chios) at three stages of lactation ranged from 0.50 to 0.68 in all samples examined by Sinanoglou et al. (2015) [100]. These values were a little bit higher than the ones reported in this research. Moreover, human milk and mare milk were shown to have the highest HH ratio. Here, this could be concluded that the diversity in these values could be associated with different breeds chosen for the research.

### 3.5. The Minerals in Milk 

The human diet is perceived to be a poor source of main minerals. However, mammals’ milk is shown to be a very good source of minerals that are crucial for the whole organism’s functioning. The insufficient supply (deficiency as well as the excess) of minerals could be the reason for the development of modern world metabolic diseases [101]. Previous studies showed the impact of an incorrect supply of minerals on the development of coronary heart disease, osteoporosis, diabetes and some types of cancer. In Poland, faulty diet and nutrition were associated with an insufficient daily supply of calcium, phosphorus, magnesium, iron, zinc, and copper [102,103].

According to previous studies, the concentration of elements obtained in mare milk was lower compared to cow milk and goat milk. Generally, the values of Ca and Mg were higher, Na and K were similar, but Cu, Fe and Zn were lower than those in human milk [104].

In this study, goat milk was characterized by the highest value of zinc (0.69 ± 0.17 mg/dL), magnesium (17.30 ± 2.70 mg/dL) and potassium (183.60 ± 17.20 mg/dL), respectively (Table 5). In Park’s (2000) studies, goat milk presented the highest values of potassium and manganese in comparison with cow and sheep milk [18].

The highest values of zinc and magnesium were reported by Raynal-Ljutovac et. al. (2008) in sheep milk [105]. It was also characterized by the highest values of calcium (200 mg/100 g), magnesium (21.0 mg/100 g) and selenium (3.1 µg/100 g). Calcium (159–242 mg/dL) and magnesium (16–25 mg/dL) were also the main minerals found in sheep milk in comparison with other mammals in Claeys et al.’s (2014) studies. In this research, sheep milk had the highest values of sodium (52.10 ± 3.2 mg/dL) and calcium (181.70 ± 17.2 mg/dL) [37].

Park’s (2000) study revealed that sheep milk presented the highest values of calcium and phosphorus in comparison with cow and goat milk. Zinc values were comparable in all analysed species. In Claeys et. al (2014) study, sheep milk presented the highest level of calcium (159–242 mg/dL), sodium (28–59 mg/dL), phosphorus (124–175 mg/dL) and zinc (0.40–0.60 mg/dL) [37].

The Singth et al. (2019) study revealed that buffalo milk is the richest source of essential minerals in comparison with the milk of other species [74].

This research also revealed that human milk was characterized by the highest concentration of iron (0.20 ± 0.10 mg/dL). This value was similar to that reported in mare milk (0.19 ± 0.10 mg/dL) (Table 5). This value was also the same as reported by Park et al. (2007), however the highest iron concentration was shown in camel milk (290 µg/100g). In Claeys et al. (2014) study, the highest iron concentrations were reported in donkeys (0.04–0.26 mg/dL) as well as in human milk (0.04–0.20 mg/dL) [37].

It should be noted here that the mineral composition of milk is not constant because it depends on the lactation phase, nutritional status of the animal, and environmental and genetic factors [106]. The influence of both lactation stage and animals’ diet on milk minerals has been shown previously in mare milk. During the milking period, a high-quality forage maintained the major mineral composition of mares’ milk in a 1.5–2 times higher amount than the milk of mares fed by pasture and a low-quality feed [107].

### 3.6. Summary

There are many factors affecting the composition of milk e.g., species, breed, and the age of animals, milk yield, as well as the following lactation, lactation phase, animal’s health status and diet. Moreover, even a minor detail such as the time of the day may have an impact on the content of some nutrients [108,109]. Additionally, the milk fat fatty acid profile is influenced by many factors, such as breed, period of lactation, nutrition, age and also the presence of environmental pollution [2,3].

These differences can make milk from various mammals a good nutritional source for human nutrition. What is more, these variations could help to choose the best food for human nutrition in the context of human diet insufficiency. As a food ingredient or consumed by itself, milk provides an excellent nutritional profile in the human diet.

What is more, some special products for human consumption have been produced on the basis of mare’s milk. The growing interest in this type of milk was aroused due to the similarity of its chemical composition to that of human milk. Mare’s milk products are very common in Russia and Central Asia throughout Mongolia [110]. Moreover, more frequent use of mare milk as a dietary supplement was also shown for elderly citizens, convalescent individuals, and mainly for children allergic to cow milk [111]. Human and equine milk are similar in terms of sugar supply (lactose and galactose), proteins and minerals, but they differ in fat content, which is markedly higher in human breastmilk [12]. For this reason, equine milk has been effective in treating children who have a cow milk protein allergy (CMPA), especially during early childhood.

This study showed that goat milk could be very good source of energy because of the high content of fat. Previously, it was also shown that, in comparison with cow milk, goat milk demonstrated nutritionally desirable traits, such as lower concentrations of C12:0, C14:0, C16:0 and Na:K ratios, and higher concentrations of cis-polyunsaturated fatty acids (PUFA), eicosapentaenoic acid (EPA), docosahexaenoic acid (DHA), isoflavones, B, Cu, Mg, Mn, P and I, although the latter may be less desirable in cases of high milk intakes. However, in contrast with nutritional targets, it had lower concentrations of omega-3 PUFA, vaccenic acid, lignans, Ca, S and Zn [112].

Previous studies on health benefits of human milk showed that cholesterol provision with breastfeeding modulates infant sterol metabolism and may induce long-term benefits. From the nutritional point of view, the knowledge about the exact ChC in different mammals’ milk will help to choose the best food for humans. Moreover, it was shown previously that the concentration of ChC in the colostrum of mammals is generally higher than in mature milk or is at a constant level. Thus, this source could be used when there is a need of higher cholesterol intake. However, no scientific studies determined the concentration of cholesterol in human colostrum, which could become a future research aim.

It was hypothesised that milk is almost a complete food for human beings, having all the essential nutrients, including minerals [113]. In this study, the desirable traits of different milk types were determined and it was assumed that the knowledge in this area could help to provide good nutritional sources for human nutrition.

## 4. Conclusions

This study compared different components of human milk and showed that milk is a good nutritional source of fat, cholesterol, macro- and microelements as well as fatty acids. Adults could consume all sorts of milk from different species. On the other hand, World Health Organisation recommends human milk as the best source of infant nutrition. It’s mainly because of the fragile gut of young organisms. It was previously shown that even infant formulae dedicated and recommended for early life does not ensure sufficient supply of desirable components (e.g., cholesterol). It was shown that the most similar to human milk is mare milk. However, none of the mammals’ milk will replace women’s milk for infant nutrition. Thus, exclusive breastfeeding is recommended. 

This is the first study comparing the concentration of most known milk components in different mammals using the same analytical methods. All milk types have specific and characteristically desirable traits. The lowest atherogenicity (AI) and thrombogenicity (TI) indices were calculated for human and mare milk, which could make these types of milk the best nutritional source for people with the risk of developing cardiovascular disease. The differences in the nutritional value of milk could be perceived as a milk profile marker, helping to choose the best food for human nutrition.

## Figures and Tables

**Table 1 nutrients-12-01404-t001:** The overall fat content in milk from different animal species based on literature data.

Animal Species	Mean Fat Content [%]	Author, Date
Human	3.64	Malacarne et al., 2002 [12]
	3.72	Butts et al., 2018 [10]
	3.80	Kamelska et al., 2013 [7]
	3.90	Thakore & Jain, 2018 [9]
Mare	1.21	Malacarne et al., 2002 [12]
	4.20	Markiewicz-Kęszycka et al., 2014 [15]
Cow	2.50	Thakore & Jain, 2018 [9]
	3.30	Gantner et al., 2015 [8]
	3.30	Balthazar et al., 2017 [16]
	3.40	Thakore & Jain, 2018 [9]
	3.61	Malacarne et al., 2002 [12]
	3.80	Paszczyk et al., 2019 [17]
Sheep	5.90	Balthazar et al., 2017 [16]
	6.90	Ferro et al., 2017 [13]
Goat	3.10	Paszczyk et al., 2019 [17]
	3.40	Park, 2000 [18]
	3.80	Balthazar et al., 2017 [16]
	4.20	Ferro et al., 2017 [13]
	5.20	Thakore & Jain, 2018 [9]
Buffalo	6.00	Thakore & Jain, 2018 [9]
	15.00	Gantner et al., 2015 [8]

**Table 2 nutrients-12-01404-t002:** Comparative cholesterol concentrations (ChCs) in different mammals’ mature milk and the analytical methods used for its determination.

Species	ChC (mg/dL)	Method	Author, Date
Cow *(Bos taurus)*	10.23–14.31 (different breeds)	Enzymatic BioAnalysis [27]	Bonczar et al., 2016 [28]
	15.71–25.39 (different breeds)	Balice Animal Husbandry Institute	Litwińczuk et al., 2014 [29]
	2.37–3.06	Cerutti, Machado, & Ribolzi, 1993	Cerutti et al., 1993 [30]
	16.71	direct saponification [31] & enzymatic method	Kamelska et al., (2015) [32]
	20.58	IDF & gas chromatography [31]	
Mare (*Equus cabalus*)	2.04	Strzałkowska et al., 2009 [33]	Markiewicz-Kęszycka et al., 2014 [15]
	2.74	saponification and reverse phase liquid chromatography [34]	Navrátilová et al., 2018 [34]
Goat (*Capra hircus*)	11.00	no data	Pandya & Ghodke 2007 [35]
	9.00–13.00	no data	Kostyra et al., 1996 [36]
	10.70–18.10	different methods	Claeys et al., 2014 [37]
Sheep (*Ovis aries*)	14.00–29.00	different mehods	Claeys et al., 2014 [37]
Yak (*Bos mutus*)	14.25	Fletouris et al., 1998 [31]	He et al., 2011 [38]
Human (*Homo sapiens*)	12.00	Park & Addis, 1986 [39]	Scopesi et al., 2002 [40]
	10.57	Paradkar & Irudayaraj, 2002 [41]	Kamelska et al., 2013 [7]
	9.88	IDF Standard	Kamelska et al., 2013 [7]
	7.05	IDF Standard	Kamelska et al., 2012 [42]
	5.38	Kamelska et al., 2012 [42]	Kamelska et al., 2012 [42]

**Table 3 nutrients-12-01404-t003:** The fat content (FAT; %), cholesterol concentration (ChC; mg/dL) and fatty acid composition of selected mammals’ milk.

Variable	Human			Cow			Mare			Goat			Sheep		
	Mean	SD		Mean	SD		Mean	SD		Mean	SD		Mean	SD	
FAT	3.53	1.02	ab	2.90	0.19	bc	1.21	0.85	c	4.14	1.29	a	7.10	3.21	a
ChC	9.90	6.51	bc	20.58	4.21	a	6.30	1.08	c	11.64	1.09	b	17.07	1.18	a
C4:0	0.02	0.03	c	3.14	0.27	a	0.18	0.09	b	2.56	0.12	a	2.81	0.14	a
C6:0	0.09	0.05	c	2.17	0.25	a	0.28	0.26	c	2.79	0.04	a	2.54	0.13	a
C8:0	0.19	0.09	d	1.41	0.17	c	2.45	1.81	ab	3.32	0.15	a	2.60	0.13	b
C10:0	1.46	0.56	c	3.25	0.54	b	6.67	3.51	b	11.28	0.69	a	9.88	0.43	a
C12:0	5.53	2.33	a	3.63	0.50	b	5.83	3.40	ab	5.62	0.77	a	6.76	0.26	a
C13:0	0.05	0.03	b	0.29	0.02	a	0.09	0.04	b	0.20	0.02	a	0.27	0.05	a
C14:0	6.40	2.79	c	11.62	1.15	b	6.37	2.12	c	11.35	0.90	b	14.98	0.37	a
C15:0	0.67	0.20	c	3.43	0.12	a	0.26	0.18	c	1.42	0.19	b	2.02	0.16	b
C16:0	25.40	3.95	a	24.90	1.40	a	22.74	2.30	a	27.69	1.39	a	29.79	0.46	a
C17:0	0.49	0.13	b	0.93	0.03	a	0.31	0.19	b	0.83	0.07	a	0.92	0.02	a
C18:0	6.14	1.29	b	12.67	1.59	b	2.14	0.31	c	8.25	0.94	a	4.77	0.27	b
C20:0	0.17	0.06	a	0.21	0.09	a	0.09	0.07	a	0.20	0.04	a	0.15	0.01	a
C10:1	0.01	0.01	b	0.28	0.02	a	0.54	0.41	a	0.30	0.02	a	0.36	0.02	a
C12:1	0.02	0.02	a	0.10	0.02	a	0.05	0.04	a	0.03	0.01	a	0.02	0.01	a
C14:1	0.23	0.10	c	0.77	0.11	a	0.21	0.11	c	0.21	0.3	c	0.41	0.01	b
C16:1	2.24	0.81	ab	1.03	0.24	b	4.45	1.47	a	1.18	0.09	b	2.06	0.01	b
C17:1	0.29	0.37	a	0.23	0,06	a	0.25	0.08	a	0.30	0.06	a	0.38	0.01	a
C18:1	40.25	8,45	a	24.81	3.81	b	25.15	3.68	b	19.77	0.56	b	15.73	1.39	c
C20:1	0.52	0.27	a	0.15	0.04	b	0.49	0.25	a	0.05	0.02	c	0.04	0.02	c
C18:2	8.84	3.68	b	2.81	0.42	c	14.94	5.75	a	2.23	0.19	c	1.97	0.55	c
CLA ^1^	0.24	0.6	b	1.59	0.70	a	0.01	0.01	c	0.51	0.11	b	1.13	0.04	a
C18:3	0.78	0.48	bc	0.86	0.09	b	7.05	2.41	a	0.23	0.08	c	0.76	0.05	b
Σ SFA ^2^	46.60	7.88	b	67.73	5.33	a	47.40	11.57	b	75.50	0.69	a	77.50	0.92	a
Σ MUFA ^3^	43.55	8.33	a	27.30	4.22	b	31.14	4.93	ab	21.83	0.52	c	19.01	1.35	c
Σ PUFA ^4^	9.85	4.13	b	5.25	1.14	b	22.01	7.57	a	2.97	0.33	c	3.86	0.49	b

^1^ CLA—conjugated linoleic acid; (% of total fatty acids); ^2^ Σ SFA—sum of saturated fatty acids (% of total fatty acids); ^3^ Σ MUFA—sum of monounsaturated fatty acids (% of total fatty acids); ^4^ Σ PUFA—sum of polyunsaturated fatty acids (% of total fatty acids); a,b,c,d—statistically significant differences.

**Table 4 nutrients-12-01404-t004:** The lipid quality indices calculated for different mammalian species.

Variable/Species	Human		Cow		Goat		Mare		Sheep	
	Mean	SD ^4^	Mean	SD	Mean	SD	Mean	SD	Mean	SD
AI ^1^	1.12	0.43	2.37	0.52	3.17	0.19	1.11	0.53	4.21	0.23
TI ^2^	0.84	0.23	1.63	0.33	2.06	0.10	0.58	0.19	2.30	0.13
HH ^3^	1.67	0.65	0.83	0.20	0.59	0.03	1.65	0.45	0.44	0.02

^1^ AI—index of atherogenicity; ^2^ TI—index of thrombogenicity; ^3^ HH—hypocholesterolaemic/hypercholesterolaemic ratio; ^4^ SD—standard deviation.

**Table 5 nutrients-12-01404-t005:** The mean and standard deviation (SD) of mineral content in selected mammals’ milk (mg/dL).

Variable/Species	Human			Cow			Mare			Goat			Sheep		
	Mean	SD ^1^		Mean	SD		Mean	SD		Mean	SD		Mean	SD	
Ca ^2^	27.6	2.9	d	119.8	12.5	b	92.9	5.8	c	130.4	10.5	b	181.7	17.2	a
Mg ^3^	3.8	1.4	c	12.6	1.9	b	8.1	3.2	bc	17.3	2.7	a	12.5	3.1	ab
K ^4^	71.3	9.0	b	147.9	19.7	a	87.1	18.3	b	183.6	17.2	a	178.6	8.4	a
Na ^5^	15.9	1.5	c	49.3	5.3	a	17.4	3.2	c	35.9	2.4	b	52.1	3.2	a
Fe ^6^	0.20	0.10	a	0.08	0.02	a	0.19	0.10	a	0.07	0.02	a	0.08	0.02	a
Zn ^7^	0.46	0.20	ab	0.62	0.18	a	0.21	0.11	b	0.69	0.17	a	0.58	0.21	a

^1^ SD—standard deviation; ^2^ Ca—calcium; ^3^ Mg—magnesium; ^4^ K—potassium; ^5^ Na—sodium; ^6^ Fe—iron; ^7^ Zn—zinc; a,b,c,d—statistically significant differences.

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
