# Peer review of "The Comparison of Nutritional Value of Human Milk with Other Mammals’ Milk"

_nutrients, 2020, doi:10.3390/nu12051404_

Round 1
Reviewer 1 Report
This is a very well designed and performed study, well written and presented. The subject is relevant and of practical importance.
The introduction widely presents current knowledge, however the data of fat content in milk are misleading as tehre are no data on the lactation period refers to (whoch day of lactation). I would suggest to add this information in the table)
Material section: there are no data on the time of material collection (which day of lactation) in human and animals. I suggest to include this information.
Author Response
We would like to thank the reviewers for careful reading of the manuscript titled: The comparison of nutritional value of human milk with other mammals’ milk. We have searched all the article for minor mistakes and changed it according to the reviewers’ comments.

Reviewer 2 Report
Interesting manuscript.
Requires supplement and correction.
Line 53 – „raindeer milk” ? Mistake, correct.. „reindeer milk”
Line 69 – triacylglicerols …correct,, triacylglycerols
Line 158-162 - Whose test results are these?
Line 173-174 – Human milk , which month of lactation?
Line 278, 282, 283- non-goat, sheep's milk has the most fat - that's how it is in Table 3. Correct.
Line 297 – correct: bovine milk
Line 301 – correct: sheep milk
Line 329 - non-goat, sheep's milk has the most SFA - that's how it is in Table 3 . Mixed up test results SFA.
Line 331 - Whose test results are these?
Line 353 -355 Did you want to write?
In this study, the lowest values of mono- and poly- unsaturated fatty acids were reported in sheep milk (19.01±1.35 % vs. 3.86±0.49 %) and goat milk (21.83±0.52 % vs. 2.97±0.33 %), respectively.
Line 382-411 You forgot to add a table with minerals. You write about table 4 (line 391) and she has no minerals..
Author Response

(The authors gave the same response as above.)

Reviewer 3 Report
Dear Authors,I really like your work and the amount of information collected! The results are interesting and worth publishing.
However, you did not avoid mistakes: you have to carefully check the data given in the tables with those published in the text e.g. in the results you state that the highest fat level was determined in goat milk (verse 278) while the table shows that it was the highest for sheep (7.1%). Also, when determining the significance of differences, what is in the text (verse 309) does not coincide with the data in Table 2. Similarly, the case is with other data. I am asking for detailed and thorough checking of all data figures in the table and comparing them with those in the text.
The number of cited literature make a big impression. However, as a horse specialist, I feel quite unsatisfied with citations concerning especially mare's milk (and there are many works on mare's milk!), but also e.g. sheep.
Another issue concerns the methodology, which, if the material is collected from women is detailed and accurate, it is cursory and general in relation to animals. It is not exactly written how milk samples were taken from females of particular species, how old the female animals were, in what conditions they were kept (I am very curious where 10 cows, 6 sheep, 6 goats and 12 cold-blooded mares were kept in identical conditions ... verse 193 especially since it was written that the samples were taken from small individual farms - it is not possible that the conditions in two different farms for different species were identical ...) exactly at what time of lactation, what breeds were these females etc. etc. The big difference in female numbers is also surprising: 18 women but sheep or goats 3 times less - only 6 ...
Please, clarify the topic of work and research. The current topic is simply too wide, too extensive, it does not reflect the nature of the work. In verse 163 it is written that no one has ever compared the nutritional value of mammalian milk, which is not entirely true because in 2006 Mr. Park and colleagues published a textbook on non-bovine mammals milk. It was based on many different researches on animals! I feel that your work is valuable and worth publishing but you have to correct some mistakes made in the text.
Author Response

(The authors gave the same response as above.)
